# Spiking Token Mixer:
# An Event-Driven Friendly Former Structure for Spiking Neural Networks

**Shikuang Deng**[1*]**, Yuhang Wu**[1*]**, Kangrui Du**[1,2]**, Shi Gu**[1,3✉]

[1]University of Electronic Science and Technology of China
[2]College of Computing, Georgia Institute of Technology
[3] Shenzhen Institute for Advanced Study, UESTC
dengsk@uestc.edu.cn, gus@uestc.edu.cn

## Abstract

Spiking neural networks (SNNs), inspired by biological processes, use spike signals for inter-layer communication, presenting an energy-efficient alternative to traditional neural networks. To realize the theoretical advantages of SNNs in energy efficiency, it is essential to deploy them onto neuromorphic chips. On clock-driven synchronous chips, employing shorter time steps can enhance energy efficiency but reduce SNN performance. Compared to the clock-driven synchronous chip, the event-driven asynchronous chip achieves much lower energy consumption but only supports some specific network operations. Recently, a series of SNN projects have achieved tremendous success, significantly improving the SNN's performance. However, event-driven asynchronous chips do not support some of the proposed structures, making it impossible to integrate these SNNs into asynchronous hardware. In response to these problems, we propose the Spiking Token Mixer (STMixer) architecture, which consists exclusively of operations supported by asynchronous scenarios, including convolutional, fully connected layers and residual paths. Our series of experiments also demonstrates that STMixer achieves performance on par with spiking transformers in synchronous scenarios with very low timesteps. This indicates its ability to achieve the same level of performance with lower power consumption in synchronous scenarios. The codes are available at https://github.com/brain-intelligence-lab/STMixer_demo.

## 1 Introduction

Spiking neural networks (SNNs) are bio-inspired neural networks that use spike signals as a carrier for inter-layer transmission. Because of the spiking mechanism, SNNs have the potential to turn high-precision multiplication into addition, which makes them a better energy-saving infrastructure [Maass, 1997, Roy et al., 2019]. However, such an advantage in efficiency is not guaranteed for typical hardware [Kim et al., 2019] but for neuromorphic chips [Akopyan et al., 2015, Davies et al., 2018b, Pei et al., 2019]. The computational energy of the synchronous neuromorphic hardware is affected by the amount of calculation, the spike frequency, and the SNN time steps [Rathi and Roy, 2020, Li et al., 2021]. And synchronous hardware provides support for nearly all matrix operations, even allowing for the execution of mixed networks that combine the ANN and SNN. In order to achieve low power consumption on synchronous hardware, SNNs must consider using

---

* Equal contribution
✉ Corresponding author

38th Conference on Neural Information Processing Systems (NeurIPS 2024).

shorter time steps. But too few time steps may potentially undermine the performance of SNNs. On the other hand, in theory, event-driven asynchronous hardware serves as an ideal implementation platform for such a mechanism. On the event-driven hardware, each neuron is configured as an independent entity driven by events rather than the high-frequency clock. Compared to synchronous scenarios, asynchronous scenarios impose more stringent constraints on network architecture. Hence, in asynchronous neuromorphic hardware, SNN usually utilizes the soft-reset Integrate-and-Fire (IF) model as the fundamental neuron model. The supported network operations in asynchronous scenarios adhere to the rule that matrix multiplication should take place between a spike matrix and a fixed float weight matrix. Consequently, due to the accumulation of membrane potential in the IF neuron, spike arrival timing errors do not significantly impact subsequent layers.

Over the past years, there have been a lot of efforts dedicated to advancing the performance of SNNs across various datasets. These efforts have explored a multitude of perspectives, aiming to optimize the training pipeline [Neftci et al., 2019, Wu et al., 2018, Li et al., 2021, Deng et al., 2023], refine the optimization objectives [Deng et al., 2021, Guo et al., 2022b], and design the network structure [Fang et al., 2021a, Zhou et al., 2023b,a]. Recently, Spikformer [Zhou et al., 2023b,a] has achieved remarkable results by incorporating the Transformer into SNN, resulting in unprecedented performance gains across a wide range of datasets. Spikformer has designed a new attention module, SSA, which aligns Spikformer with the forward rules of SNN during the inference phase. Such progression in a Spikformer-like design theoretically eliminates the need for high-precision matrix multiplication. Spikformer-like models have indeed demonstrated significant performance improvements in synchronous hardware or GPU simulation. However, an easily overlooked point is their feasibility in implementing asynchronous hardware. In event-driven scenarios, the spike arrival times may not be as precise as in synchronous scenarios with hardware clocks. Calculating two spike matrix multiplications or a spike matrix passing through the max-pooling layer could result in significant differences in the output.

To address these issues, we proposed a new architecture called Spiking Token Mixer (STMixer), which only uses asynchronous neuromorphic hardware-compatible operations, including convolutional, fully connected layers and residual paths. STMixer still adopts the holistic framework of spikformer, which consists of spiking patch splitting (SPS), encoders, and classification heads (Fig. 1 (A)). The encoder module in STMixer consists of two components: the token mixing module and the channel module. In the token mixer component of STMixer, the value matrix undergoes matrix multiplication with a trainable attention map weight matrix (Fig. 2 (A)). This new architecture eliminates the requirement of performing matrix multiplication between two spike matrices during the forward pass, thereby conforming to the event-driven situation. We also analyzed and enhanced the SPS module, ensuring its compliance with the requirements of an asynchronous environment while reducing information loss. Furthermore, we employed the proposed mixer architecture to optimize the components of the surrogate module [Deng et al., 2023] and used the surrogate module learning to assist the SNN training. The following summarizes our main contributions:

- We examine problems with the SSA module in Spikformer in asynchronous scenarios and suggest a new module, the Spiking Token Mixing (STM) module, which consists solely of network components that cater to asynchronous environments.

- We investigate the SPS module's asynchronous support and the information loss problem. To address these concerns, we proposed the information protection spiking patch splitting (IPSPS) module, which removes the maxpool layer and adds a residual path to protect information.

- Through the network architecture search, we verified that the STMixer structure excels among all possible configurations in terms of performance and parameter quantity. Additionally, we have validated that STMixer achieves performance on par with or even surpasses existing Spikformer-like works in synchronous scenarios by using only a single time step across almost all datasets. This suggests that STMixer has the potential to deliver high performance with lower power consumption in synchronous scenarios, too.

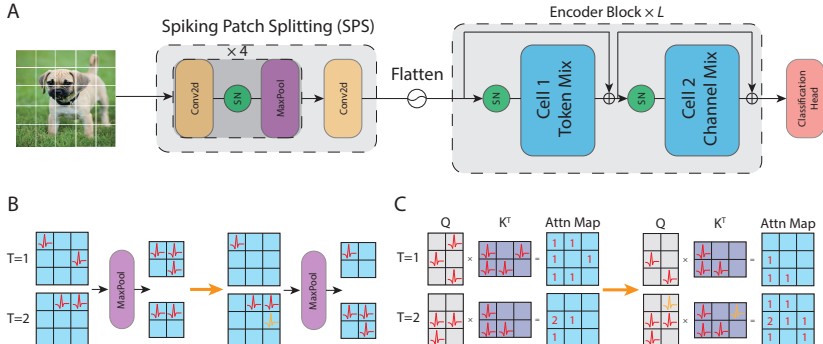

Figure 1: The architecture of Spikformer-like networks and the issue of spiking matrix multiplication and max pooling layers (A) The overall architecture of STMixer and Spikformer. (B) The issue of the Max Pooling layer when there is a spike delay in asynchronous hardware. (C) The spike delay issue of spiking matrix multiplication. When both matrices involved in matrix multiplication are composed of spikes, the imprecise timing of spike arrival can result in inaccuracies in the computed product.

## 2 Related Work

### 2.1 Transformer-like architectures in artificial neural networks

Convolutional networks, such as ResNet and VGG, have historically been the most popular architectures in the realm of computer vision [Zheng et al., 2021, Rathi and Roy, 2020]. Some works have tried to add attention mechanisms to visual tasks [Yao et al., 2023, Zhu et al., 2024] after being inspired by the amazing results of transformers in NLP [Vaswani et al., 2017, Devlin et al., 2019, Floridi and Chiriatti, 2020]. Especially, the Vision Transformer (VIT) [Dosovitskiy et al., 2020] has demonstrated that transformer architectures can achieve remarkable performance in computer vision tasks. VIT uses patch embedding as an input and multiple stacked transformer encoders as the backbone. Since then, a lot of research has been done to improve performance and efficiency by making changes to how images are stored [Parmar et al., 2018, Chen et al., 2020, Liu et al., 2021], how attention is calculated [Shen et al., 2021, Bolya et al., 2022], and other methods [Touvron et al., 2021, Zhang and Sabuncu, 2020]. Meanwhile, some works have revealed that just employing MLPs as the token mixer can also lead to good performance[Tolstikhin et al., 2021, Touvron et al., 2022, Guo et al., 2022a]. Recently, a fresh perspective has been proposed by Metaformer[Yu et al., 2022], suggesting that the success of Transformer in visual tasks stems from the holistic encoder architecture, where token mixer and channel mixer interact. Interestingly, Metaformer even achieved competitive performance by employing a simple pooling layer as the token mixer. Their findings offer us novel insights into how to modify Spikformer.

### 2.2 Spiking neural networks

The forward process of SNN encompasses the following rule: all high-precision matrix multiplications should be capable of degrading to matrix additions, eliminating the need for high-precision matrix multiplication operations after implanting in hardware [DeBole et al., 2019, Davies et al., 2018b]. However, this rule restricts the flexibility of SNNs to incorporate various operations, limiting them to a few substructures such as the convolution layer, the fully connected layer, the pooling layer, and so on. The majority of SNN architectural designs are indeed based on these substructures, such as the advancements of ResNet structures like SEW-ResNet [Fang et al., 2021a] and MS-ResNet [Hu et al., 2024], as well as the structure search efforts like AutoSNN [Na et al., 2022] and SNASNet [Kim et al., 2022]. Recently, Spikformer [Zhou et al., 2023b] has achieved remarkable success by integrating the transformer architecture into SNNs, attaining SOTA performance across various datasets. The authors proposed the SSA module to ingeniously realize attention computation in SNN while adhering to the aforementioned forward rule. At present, Spikformer and its subsequent series [Zhou et al., 2023a, Yao et al., 2024] of improvements are steadily pushing the boundaries of SNN performance. Nevertheless, though SSA exhibits notable performance in clock-driven hardware, it

may encounter significant discrepancies in event-driven hardware due to the fact that the spikes may not arrive at the same time.

## 3 Preliminaries

### 3.1 Spiking neural model

The spike neuron serves as the fundamental unit of SNN, wherein their somatics harbor membrane potential that integrates information from the synaptic according to prescribed model principles. The leaky integrate-and-fire (LIF) model [Lapicque, 1907] is currently the most prevalent SNN neuron model, which describes the membrane potential update rule as follows:

$$
\begin{aligned}
H[t] &= \frac{1}{\tau}V[t-1] + X[t] \\
S[t] &= \Theta\left(H[t] - V_{\text{th}}\right), \\
V[t] &= H[t](1 - S[t]) + V_{\text{reset}}\,S[t],
\end{aligned}
\tag{1}
$$

where $V[t]$ is the membrane potential, $\tau$ is the membrane time constant, and $S[t]$ is the output in time $t$, it would be equal to 1 or 0, which means if there is a spike or not. $\Theta(\cdot)$ means the Heaviside step function, which equals 1 when $H[t]$ exceeds the threshold value $V_{\text{th}}$. Furthermore, we employ the hard reset mechanism, which means the membrane potential will reset to the reset potential when the neuron fires a spike.

### 3.2 Spikformer

We use the Spikingformer [Zhou et al., 2023a] architecture as the foundational framework of our network. Spikingformer addresses the issue of residual connections in Spikformer by using membrane potential residual instead of spiking residual. The overall structure of Spikingformer, as depicted in Fig. 1 (A), comprises three main components: Spiking Patch Splitting (SPS), an encoder module, and a classification head. The SPS consists of five convolutional layers, where the first four layers are followed by a max pooling layer. The function of SPS is to perform convolution processing and downsampling on input images, resulting in an output matrix with the shape $D \times h \times w$. Eventually, this matrix is flattened into $N$ ($N = h \times w$) patches, each with a $D$-dimensional vector. The classification head consists of an adaptive average pooling layer, which is used for downsampling, and a fully connected layer, responsible for obtaining the network's final output. The classification head will introduce floating-point matrix multiplication, so it needs to be deployed on the CPU. Because the calculation amount of the classification head is very small, it will not cause much energy overhead or inference delay but will greatly improve SNN performance [Zhou et al., 2023a].

The most important component of Spikformer is its encoder module, which consists of two cells and residual connections (Fig. 1 (A)). Cell 2 corresponds to the Feed Forward Network (FFN) in VIT. It consists of two fully connected layers designed for channel mixing. Cell 1 refers to the Spiking Self-Attention (SSA) module proposed by the Spikformer. SSA has been demonstrated to be more suitable for SNN compared to vanilla self-attention (VSA). In SSA, when the input is $X \in \mathbb{R}^{N \times D}$, the query ($Q \in \mathbb{R}^{N \times D}$), key ($K \in \mathbb{R}^{N \times D}$), and value ($V \in \mathbb{R}^{N \times D}$) are computed using the following equations:

$$
Q = \mathcal{SN}_Q\left(\text{BN}\left(XW_Q\right)\right), K = \mathcal{SN}_K\left(\text{BN}\left(XW_K\right)\right), V = \mathcal{SN}_V\left(\text{BN}\left(XW_V\right)\right)
\tag{2}
$$

where $W_Q, W_K, W_V \in \mathbb{R}^{D \times D}$ are learnable weight matrices, $\mathcal{SN}$ means spiking neural (LIF model) activation, and BN denotes the batch normalization layer. The authors of Spikformer argue that in ANN, the presence of outliers or negative values in the query and key makes the attention map require a softmax function. However, since the output of eqn. 2 is a binary matrix that only consists of 0 and 1, there are no outliers or negative values. Therefore, they directly perform matrix multiplication on the three matrices to obtain the output of SSA:

$$
\text{SSA}(Q, K, V) = \mathcal{SN}\left(QK^{\text{T}}V * s\right)
\tag{3}
$$

where $s = 0.125$ is a constant factor.

# 4 Methodology

## 4.1 Event-driven Scenarios

In ideal asynchronous scenarios, each neuron is an autonomous entity solely focused on updating its membrane potential upon receiving spike signals, then firing a spike once the membrane potential exceeds the threshold. Since the model in asynchronous scenarios lacks the specific hardware clock to control the spike firing and arriving time, it is inevitable to encounter temporal errors in the spike arrival time. In the extreme case, it is conceivable to assume that each neuron fires spikes at an individual time (the SNN time step T approaches infinity, and each spike fires at an individual time step). When there are errors in the arrival time of spikes ($t = t + \sigma$), the output sum of asynchronous scenarios supported operations like convolution, fully connected layer, and average pooling will not change. However, some operations that are commonly observed in synchronous scenarios may have output errors due to input spike arrival timing error.

**Maxpool layer.** As shown in Fig. 1 (B), a spike at a specific location arrives one time step later in the max pooling layer, resulting in a noticeable change in the output. This output error will further affect the result of the subsequent convolution layer, as well as the membrane potential accumulation of neurons and their output spikes. In the extreme case where each spike has an independent arrival time, all the spikes will pass through the max pooling layer. Therefore, the max pooling layer needs a clock to control the spike arrival time and cannot be simply deployed in event-driven scenarios. To address this problem, we can simply remove the max pooling layer and change the stride of the convolutional layer following the max pooling layer to 2 for downsampling. After this modification, when the arrival spikes have temporal discrepancies, the cumulative membrane potential of the post-convolution spike neurons will remain relatively stable.

**Spike matrix multiplication.** On the synchronization and clock-driven hardware, the query, key, and value spike matrices can arrive on time. This guarantees that the forward SSA results on this hardware correspond to the GPU simulation results. However, on event-driven hardware, the absence of clock constraints often makes it challenging for spikes to arrive at precise timings. If there is any spike loss or similar occurrence during matrix multiplication in SSA, a significant error in the SSA output will occur. Here is a simple example that shows how the computed attention map matrix changes significantly in SSA when there is a spike delay at a certain point in both the query and key matrices (Fig. 1 (C)). This example shows how hard it is to make sure that the forward process with two spike matrix multiplications runs smoothly on event-driven neuromorphic hardware.

## 4.2 Clock-driven Scenarios

According to Rathi and Roy [2020], Hu et al. [2024], Zhou et al. [2023a], the theoretical computational energy consumption of SNNs on clock-driven scenarios can be defined as:

$$E_{\text{SNN}} = E_{AC} \times \sum_{l=2}^{L-1} SOP_{Conv}^l + E_{MAC} \times (FLOP_{Conv}^1 + FLOP_{Linear}^L), \qquad (4)$$

where $E_{AC} = 0.9pJ$ and $E_{MAC} = 4.6pJ$ are the assumed energy consumption of accumulate and multiply-and-accumulate operations, respectively, of 45nm CMOS hardware [Horowitz, 2014], $SOP_{Conv}^l$ denotes the number of spike-based accumulate operations, and $FLOP_{Conv}^1$ means the computation of the first layer that encodes static RGB images into a spike matrix [Zhou et al., 2023a], $FLOP_{Linear}^L$ means the computation of the classification head. And $SOP_{Conv}^l = FLOP_{Conv}^l \cdot fr^l \cdot T$ is affected by the SNN fire rate $fr^l$ and time step $T$. However, recent works [Dampfhoffer et al., 2022, Bhattacharjee et al., 2024] have pointed out that Eqn. 4.2 neglects the overhead of memory scheduling $E_{Mem}$, which is much larger than $E_{MAC}$.

On each time step of SNN forward inference, hardware will deal with memory scheduling, so $E_{MAC}$ is also affected by the time step $T$. And SNNs with a large number of time steps may have a large energy overhead in synchronous scenarios, even potentially surpassing that of ANNs under the same structure. Therefore, we must closely examine the performance of the new SNN architecture at extremely low time steps to ensure its adaptability to low-power scenarios.

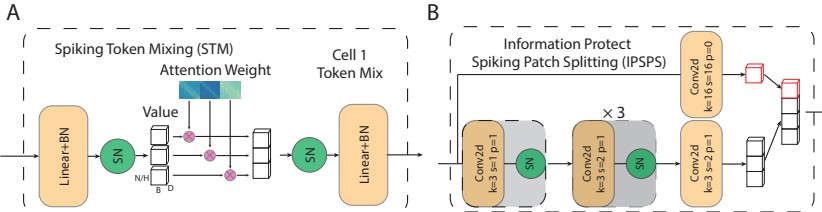

Figure 2: The architecture of (A) Spiking token mixing (STM) module and (B) Information protect Spiking Patch Splitting (IPSPS) module.

## 4.3 STMixer

To address the aforementioned requirements in the two scenarios, we present the STMixer network architecture. STMixer solely consists of asynchronously supported operations and performs well at extremely low time steps ($T = 1$). STMixer has the same overall structure as Spikformer but uses STM as the token mixing module and IPSPS to replace the SPS module (Fig. 1 (A)). STMixer is controlled by three hyper-parameters: $L$, $D$, and $H$, which respectively control the number of encoders, the encoding dimension, and the number of multi-heads of STM. We use the naming convention "STMixer-L-D-H" to represent the three parameters, which is similar to Spikformer-like works. Next, we will go into detail about the important modules in STMixer.

### 4.3.1 Spiking Token Mixing (STM)

As shown in Fig.2 (A), we use the STM module to realize the token mixing function. This module utilizes a weight matrix ($W_{\text{STM}} \in \mathbb{R}^{N \times N}$) to mix the token dimension information. Mathematically, its forward process can be expressed as follows:

$$\text{STM}(X) = \mathcal{SN}(W_{\text{STM}}V) = \mathcal{SN}(W_{\text{STM}}\mathcal{SN}(\text{BN}(XW_V))), \tag{5}$$

Where $X \in \mathbb{R}^{N \times D}$ is the input spiking matrix of STM, $V \in \mathbb{R}^{N \times D}$ is the value matrix, and $W_V \in \mathbb{R}^{D \times D}$ is the projection matrix to acquire the value matrix $V$. When we simplify the operations of the SSA module by removing the activations for query and key, the formula for SSA can be simplified as follows:

$$\begin{aligned}
\text{SSA}(Q, K, V) &\to \mathcal{SN}(XW_Q W_K^T X^T V * s) \\
&= \mathcal{SN}(XWX^T V * s).
\end{aligned} \tag{6}$$

After this simplification, $W_Q$ and $W_K$ can converge into a single matrix, and the attention map is reduced to $XWX^T$. So the objective of $W_{\text{STM}}$ is to gradually fit the attention map during the training process. And some studies also suggest that linear token mixing can capture external attention between samples in the dataset [Guo et al., 2022a].

Furthermore, we evenly divide the value matrix $V \in \mathbb{R}^{N \times D}$ into $H$ groups based on dimension D, allocating a weight matrix $W_{\text{STM}}^h \in \mathbb{R}^{N \times N}$ to each group. This method employs different attention weights for various patch encoding groups $V^h \in \mathbb{R}^{N \times D/H}$. It facilitates a finer granularity in token mixing, leading to enhanced information mixing and integration among tokens. In addition, due to the relatively modest number of tokens in the vision task, implementing multi-head in this manner would not significantly increase the parameters and maintain the same level of computational complexity.

### 4.3.2 Information Loss in Spiking Patch Splitting

In a neural network, as the input $X$ undergoes successive forward propagation layers, the inherent information gradually dissipates. Mathematically this phenomenon can be described as:

$$I(X, X) \geq I(X, f(X)) \geq I(X, g(f(X))) \cdots \geq I(X, Y), \tag{7}$$

where $I$ means the mutual information, $f$ and $g$ are the operations of network layers, and $Y$ is the network output. An SNN layer may lose more information than an ANN layer because the layer output is a spike matrix, which inherently contains significantly less information than a floating-point

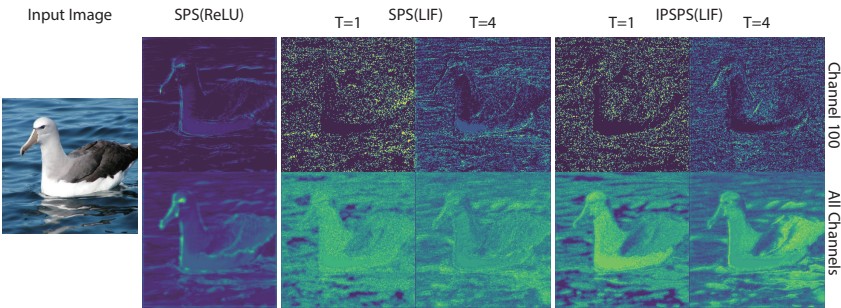

Figure 3: Visualization results of random initial weight output feature maps of the SPS module. We present the feature maps of the 100th channel and the average of all channels, and in SNN, the feature map is the mean feature map across all time steps. When using the LIF neuron model as the activation function for SPS, it is obvious that there is a significant information loss in the output feature map compared to the input image. Employing the IPSPS can alleviate this information loss.

matrix. The Spiking Patch Splitting (SPS) module comprises five layers of convolutional operations and spiking neural activation. After passing through a total of five convolutional layers and spiking neural, the feature map may have already lost a significant amount of information contained in the input images (Fig. 3).

Since Spikformer-like SNNs rely on convolutional layers in the SPS module for information extraction, reducing the number of layers in the SPS module will compromise the SNN's ultimate performance. To address this information loss problem, we add an additional path that directly uses a convolutional layer for input image encoding (Fig. 2 (B)). In our configuration, the encoding outputs from the auxiliary pathway account for $1/8$ of the entire SPS output feature map. This new encoding module is called information protected spiking patch splitting (IPSPS).

There are two advantages to IPSPS: (1) IPSPS does not reduce the convolutional layers in the original pathway; it only decreases the number of feature map channels by $1/8$ for the last convolutional layer. Theoretically, this approach should not significantly impair the SNN performance. Simultaneously, the new pathway can provide feature maps with varying convolutional depths for subsequent encoders, enriching the SPS output's feature representation. (2) The output of the additional pathway in IPSPS can be regarded as a linear transformation applied to the input image. The input feature map for the first encoder from the additional pathway is essentially the linearly transformed input image passed through an SNN neural layer, which maximally preserves the input image's information.

### 4.4 STM on surrogate module learning

Even if we reduce the information loss of the encoder's input, the accumulation of gradient errors caused by surrogate gradient may still prevent SNN from achieving better performance with low time steps. As a result, we use the surrogate module learning (SML) method [Deng et al., 2023]. The SML is an exceptionally effective approach proposed to alleviate the issue of gradient errors in SNN by adding an auxiliary pathway during the training phase. However, the use of large kernel convolutions results in significant computational overhead during the training phase. In this paper, we employ the STMixer encoder to replace the computationally intensive convolutional encoder, thereby greatly reducing the computational overhead of SML. The STM in the surrogate module uses the GeLU activation function and employs the same number of heads as the STM in the main structure. Finally, we use a classification head for the output. Starting from index 0, we place a surrogate module after every two encoders for surrogate module learning. And the number of encoders in the surrogate module is half of the count of encoders after the place point in the main pathway.

## 5 Experiments

### 5.1 Comparison to exiting works

**CIFAR.** In IPSPS, we only set the stride to 2 for the last two convolutional layers, similar to Spikformer. This implies that IPSPS downsamples the input images to a resolution of $8 \times 8$ and

subsequently obtains 64 patches through flattening. Here, STM in STMixer uses 32 heads. And we use the same training script as Spikeformer, the learning rate is set to 0.005, the batch size is 128, and training total 300 epochs. As shown in Tab. 1, STMixer achieves state-of-the-art performance both on the CIFAR-100 dataset and considering models with comparable parameter numbers and flops. However, on the CIFAR-10 dataset, the SML method slightly outperforms the results reported by us. This could be attributed to the larger parameter count of the structure they used or the structure being more suitable for the CIFAR-10 dataset. These results demonstrate the potential of STMixer to achieve excellent performance for SNN on simple static datasets, even with only one time step. It also suggests that STMixer can serve as a competitive alternative to Spikformer-like works.

**ImageNet.** The IPSPS module of STMixer downsamples the input images of 224×224 resolution to 16×16 through four successive convolutional layers, ultimately obtaining 196 patches. We employ the same training script as Spikeformer, set the batch size to 256, and set the training total epochs to 300. All the STMixers use 16 heads for STM in ImageNet experiments. We set the time step to 1, aiming to verify its performance under low energy consumption. The energy consumption of synchronous hardware estimation is computed by Eqn. 4.2. As shown in Tab. 1, we even use a much lower time step (much lower energy consumption), our STMixer-8-768-16 performs better than Spikeformer and achieves SOTA accuracy on ImageNet. This observation highlights the substantial potential of STMixer in synchronous and low energy scenarios.

Table 1: Performance comparison of our method with existing methods on CIFAR10/100 and ImageNet. Param means the number of parameters.

| Dataset | Methods | Architecture | Param (M) | Time Steps | Energy (mJ) | Accuracy (%) |
|---|---|---|---|---|---|---|
| CIFAR-10 | tdBN [Zheng et al., 2021] | ResNet-19 | 12.57 | 4 | - | 92.92 |
| | TET [Deng et al., 2021] | ResNet-19 | 12.57 | 4 | - | 94.44±0.07 |
| | AutoSNN [Na et al., 2022] | AutoSNN (C=128) | 21 | 8 | - | 93.15 |
| | SNASNet [Kim et al., 2022] | SNASNet-BW | - | 8 | - | 94.12±0.25 |
| | SpikeDHS$^D$ [Che et al., 2022] | SpikeDHS-CLA (n3s1) | 14 | 6 | - | 95.36±0.01 |
| | SML [Deng et al., 2023] | ResNet-18 | 11.22 | 4 | - | **96.04**±0.10 |
| | Spikformer [Zhou et al., 2023b] | Spikformer-4-384 | 9.32 | 4 | - | 95.19 |
| | Spikingformer [Zhou et al., 2023a] | Spikingformer-4-384 | 9.32 | 4 | - | 95.61 |
| | **ours** | STMixer-4-384-32 | 8.29 | 1 | - | **95.49**±0.13 |
| | | | | 4 | - | **96.01**±0.11 |
| CIFAR-100 | tdBN [Zheng et al., 2021] | ResNet-19 | 12.57 | 4 | - | 70.86 |
| | TET [Deng et al., 2021] | ResNet-19 | 12.57 | 4 | - | 74.47±0.28 |
| | AutoSNN [Na et al., 2022] | AutoSNN (C=128) | 21 | 8 | - | 69.16 |
| | SNASNet [Kim et al., 2022] | SNASNet-BW | - | 8 | - | 73.04±0.36 |
| | SpikeDHS$^D$ [Che et al., 2022] | SpikeDHS-CLA (n3s1) | 14 | 6 | - | 76.25±0.10 |
| | SML [Deng et al., 2023] | ResNet-18 | 11.22 | 4 | - | 79.49±0.11 |
| | Spikformer [Zhou et al., 2023b] | Spikformer-4-384 | 9.32 | 4 | - | 77.86 |
| | Spikingformer [Zhou et al., 2023a] | Spikingformer-4-384 | 9.32 | 4 | - | 79.09 |
| | **ours** | STMixer-4-384-32 | 8.29 | 1 | - | **80.00**±0.21 |
| | | | | 4 | - | **81.87**±0.16 |
| ImageNet | tdBN [Zheng et al., 2021] | ResNet-34 | 21.79 | 6 | - | 63.72 |
| | SEW-ResNet [Fang et al., 2022] | SEW-ResNet34 | 21.79 | 4 | - | 67.04 |
| | TET [Deng et al., 2021] | SEW-ResNet34 | 21.79 | 4 | - | 68.00 |
| | SpikeDHS$^D$ [Che et al., 2022] | SpikeDHS-CLA-large | 58M | 6 | - | 68.64 |
| | SML [Deng et al., 2023] | ResNet-34 | 21.79 | 4 | - | 68.25 |
| | **Spikformer [Zhou et al., 2023b]** | Spikformer-6-512 | 23.37 | 4 | 9.41 | 72.46 |
| | | Spikformer-8-512 | 29.68 | 4 | 11.57 | 73.38 |
| | | Spikformer-10-512 | 36.01 | 4 | 13.89 | 73.68 |
| | **Spikingformer [Zhou et al., 2023a]** | Spikingformer-8-512 | 29.68 | 4 | 7.46 | 74.79 |
| | | Spikingformer-8-768 | 66.34 | 4 | 13.68 | 75.85 |
| | **ours** | STMixer-6-512-16 | 23.63 | 1 | 1.96 | 73.31 |
| | | STMixer-8-512-16 | 30.12 | 1 | 2.20 | 73.82 |
| | | STMixer-10-512-16 | 36.61 | 1 | 2.48 | 74.91 |
| | | STMixer-8-768-16 | 61.16 | 1 | 4.45 | **76.68** |

## 5.2 Exploration of the architecture of Encoders

In this section, we endeavor to validate the effect of different encoder structures (the structures of Cell 1 and Cell 2 in Fig. 1). Additionally, we also want to figure out whether the metaformer-like structure is a superior architecture in SNN. As depicted in Fig. 4, each cell consists of three positions, with each position having five possible structures (STM and SSA only use a single head). Disregarding any duplicates generated by the identity structure, there are a total of $5^6$ different substructures. Conducting a comprehensive validation of all these cases is impractical. Inspired by the concept of a one-shot NAS, we train a supernet that encompasses all possible substructures. The supernet is composed of three main components: the SPS, the classification head, and four identical encoders. Each encoder consists of two cells that are used for network search. This supernet enables us to efficiently evaluate the performance of each substructure. The CIFAR training dataset is divided

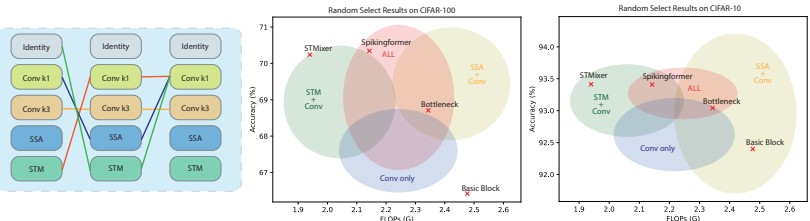

Figure 4: Neural architecture search space and search results. Left: Search space of each Cell. Mid: random sample result from supernet on CIFAR-100. Right: random sample result from supernet on CIFAR-10.

Table 2: Effect of STMixer componets and trainig methods

| Modification | Accuracy (%) |
|---|---|
| STMixer | 79.55 |
| IPSPS →SPS | 78.97 |
| STM→SSA | 78.33 |
| SML(STM)→SML(Conv) | 79.07 |
| SML→SDT | 77.17 |

Table 3: Effect of number of STM heads

| H | Accuracy (%) |
|---|---|
| 1 | 79.02 |
| 2 | 79.07 |
| 4 | 79.39 |
| 8 | 79.51 |
| 16 | 79.55 |
| 32 | 80.15 |

into training and validation sets in a ratio of 8.5:1.5. During each forward pass, the supernet will randomly select a path and train on the training set for 600 epochs. After the training is completed, we randomly sample 4000 distinct subnetworks from the supernet and evaluate their performance on the validation set. Additionally, we calculate the forward FLOPs for each of these subnetworks. Note that the FLOPs mentioned here refer to the estimated floating point operations on the training platform. The actual theoretical synaptic operations (SOPs) rely on FLOPs, the simulation time step, and spike frequency. We have categorized the subnetwork into four classes: (1) only convolutional layers; (2) combining SSA and convolutional layers; (3) incorporating STMixer and convolutional layers; and (4) simultaneously having STMixer and SSA. To draw ellipses, we use the mean and variance of accuracy and FLOPs, showcasing the relative merits of four different selection spaces. Furthermore, we have marked the positions of STMixer and Spikingformer. As shown in Fig. 4, considering the computational complexity and network performance, as well as the suitability of the components used in asynchronous scenarios, STMixer appears to be the most promising architecture (STMixer is located at the top-left corner of the result figure).

## 5.3 Ablation studies

To understand the effect of various components and training methods in STMixer, we conducted ablation experiments on CIFAR-100 with STMixer-4-384-16 ($T = 1$). The summarized results are presented in Table 2. As we can see, SML is the most significant improvement method. Considering that STMixer-4-384-16 is indeed a 25-layer network, the training process affects the nonnegligible gradient error accumulation. SML is an effective approach to mitigating the gradient error problem. And the employment of the modified SML algorithm with STM modules enables a more efficient facilitation of training SNNs (+0.48%). Then, under the SML training method, the performance of the STM module surpasses that of the SSA module (+1.22%). Finally, by preserving more information from the input images by IPSPS, the SNN achieves better performance (+0.58%).

Another parameter that significantly influences the STMixer performance is $H$, the number of heads in STM. Here, we increment the value of H from 1 to 32 and evaluate the performance of STMixer-4-384 on the CIFAR-100 dataset. As the number of heads ($H$) increases, the STMixer accuracy on the test set has improved from 79.02% to 80.15%. This observation highlights the necessity of performing token mixing at a finer granularity.

## 5.4 Benchmark on fully event-driven asynchronous scenario

Besides the superior performance demonstrated on the aforementioned static datasets, our proposed STMixer model also exhibits significant adaptability for event-driven scenarios. In these scenarios,

SNNs are deployed on asynchronous chips to receive and process event streams from DVS cameras, thereby forming a system that integrates sensing and computation. This integration leverages the sparse nature of spike activity, allowing SNNs to achieve ultra-low energy consumption compared to their ANN counterparts. To validate this advantage, we conducted additional experiments on fully asynchronous frameworks. Since the current commercial asynchronous chips (e.g., Speck of the Synsense [Richter et al., 2024] and Loihi [Davies et al., 2018a, Lines et al., 2018] of Intel) are limited in capacities and lack support for various operators such as shortcuts, we tested our models on a home-made C++ simulator that reproduces the computational principles of the Speck DevKits (see Synsense Sinabs).

Table 4: STMixer vs Spikformer under large T and event-driven scenario on CIFAR10-DVS.

| Model | w/ bias | T=25 | T=40 | T=80 | T=160 | T=320 | event-driven |
|---|---|---|---|---|---|---|---|
| STMixer(T=25) | 66.73 | 64.31 | 65.32 | 63.91 | 61.79 | 58.37 | 50.70 |
| Spikformer(T=25) | 64.21 | 56.05 | 53.83 | 34.58 | 22.78 | 19.15 | 16.60 |

| Model | w/ bias | T=40 | T=80 | T=160 | T=320 | T=640 | event-driven |
|---|---|---|---|---|---|---|---|
| STMixer(T=40) | 75.91 | 71.77 | 71.57 | 70.77 | 67.74 | 64.62 | 54.10 |
| Spikformer(T=40) | 67.74 | 10.08 | 10.08 | 10.08 | 10.08 | 10.08 | 10.00 |

We compared STMixer to a structurally identical SpikFormer, created by replacing the STM component in STMixer with SSA on CIFAR10-DVS. Due to the prohibition of clock-driven bias in a fully event-driven scenario, we implemented a two-phase training pipeline: initially training a standard SNN, then absorbing BN layers and eliminating all bias terms through additional fine-tuning, while the side effect is a slight performance degradation (see the w/ bias column and T=25(40) column in Table. 4). The training of SpikFormer becomes extremely unstable during the bias-removal phase when T=40 and fails to converge even with the learning rate reduced to $10^{-5}$.

We first deployed the two models on our event-driven simulator and evaluated their test accuracy. As shown in Table. 4) event-driven column, Spikformer undergoes significant performance degradation after deployment because events fail to meet each other and end in SSA, whereas STMixer maintains high performance and demonstrates strong compatibility with the event-driven framework. We further assessed their event-driven adaptability by inference performance at large T. When T is considerably large, events rarely coincide within the same frame, effectively approximating an event-driven scenario. According to Table. 4), as T increases, Spikformer's performance deteriorates sharply, whereas STMixer exhibits only a modest accuracy decrease. These results are consistent with our analysis in Section 4.1. Further details are provided in the appendix.

# 6 Conclusion

The advantage of spiking neural networks over artificial neural networks is their efficient and low-power inferencing capabilities once implanted in neuromorphic hardware. Within neuromorphic hardware, synchronous hardware uses a hardware clock to control when the spikes fire and arrive. The power consumption of such hardware is influenced by the time step length (T), whereby a smaller T typically results in lower power consumption. Asynchronous hardware has a lower energy consumption potential than synchronous hardware, but it has limitations on the network structure and only supports some specific matrix operations due to the lack of the hardware clock. For the aforementioned reasons, SNN must use asynchronously supported operators to build new architectures (for asynchronous hardware) or complete the performance challenge with an extremely low time step (for synchronous hardware). In this paper, we propose a new architecture named STMixer that satisfies the above challenges. STMixer is completely composed of asynchronous scenario supported components, including convolutional layers, fully connected layers, and residual paths, and has excellent performance when the time step $T = 1$. Our work will provide ideas for the SNN architecture, which works on both event-driven friendly and clock-driven efficient SNNs.

# 7 Acknowledgment

This project is supported by NSFC Key Program 62236009, Shenzhen Fundamental Research Program (General Program) JCYJ 20210324140807019, NSFC General Program 61876032, and Key Laboratory of Data Intelligence and Cognitive Computing, Longhua District, Shenzhen.

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

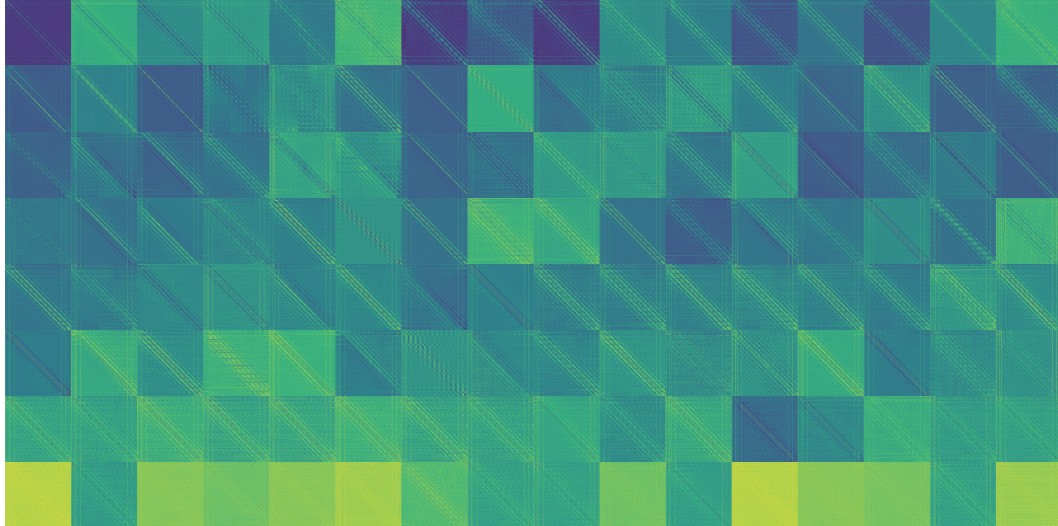

Figure 5: The STM attention weights of different encoders. Each row illustrates the 16 attention weights for each encoder.

# A Attention Weight Visualization

Here we show the token mixing attention weights of the encoders from the well-trained STMixer-8-768-16 on the ImageNet dataset. These weights provide insights into the blending process of tokens, revealing how they are mixing information. As shown in Fig. 5, most of the weight matrices exhibit a pronounced regularity: there are some parallel lines near the symmetry axis. This observation implies two insights. While the attention weight of the STM enables the extraction of global information, it still tends to favor the extraction of local information. In visual tasks, the correlation between each patch and its neighboring patches is maximized. Moreover, it can be observed that the 16 attention weight matrices in each row are different, indicating that each head of the multi-head STM is capable of extracting distinct token mixing patterns.

# B Experiments

## B.1 Experiments settings

**Training for comparison with existing works** All experiments on CIFAR were conducted on two 4090 GPUs, while experiments on ImageNet were performed on four 4090 GPUs. We employed the same training script as Spikingformer [Zhou et al., 2023a] and employed identical data augmentation techniques. In the CIFAR experiments, we modify the learning rate to be 5e-3 and the minimum learning rate to 1e-7. In the ImageNet experiments, we modify the learning rate to be 3e-4 and the minimum learning rate to 1e-5.

**Training for event-driven experiments** Models for event-driven experiments are all trained by our two-phase training pipeline. Specifically, we first train an SNN with bias and BN layers. Before further fine-tuning, we absorb BN layers and set all bias terms to 0. Finally, we fine-tune the model for more epochs with bias frozen. For the first phase, we use the AdamW optimizer and set the learning rate to 0.001 and weight decay to 0.02. For the second phase, we use the AdamW optimizer and set the learning rate to 0.001 and the weight decay to 0.005. For Spikformer and STMixer with T=25, a time step of T=10 is used for the first phase over 300 epochs. T is then set to 25 for the bias-removal phase, during which models are fine-tuned for an additional 30 epochs. For Spikformer and STMixer with T=40, a consistent T=40 is used throughout both phases. Models are trained for 100 epochs during the first phase, followed by 30 epochs of fine-tuning during the bias-removal phase. Notably, we found that MultiSpike IF neurons with soft reset (see Synsense Sinabs) are optimal for training models intended for event-driven deployment, and this configuration was applied in all event-driven experiments.

**Backbone for event-driven experiments** We employed the modified STMixer-2-256-8 as the backbone for all event-driven experiments, incorporating several adjustments to enhance its suitability for fully event-driven deployment. First, the non-spiking membrane shortcut was replaced with a standard spiking shortcut [Zheng et al., 2021] for event-driven implementation. Second, the first two convolution layers of the SPS module are replaced with a single $4 \times 4$ Avgpool layer. Building upon this foundation, we replace the STM module with SSA to construct the Spikformer model.

## B.2 Compare with existing works on neuromorphic datasets in synchronous scenarios

Table 5: Performance comparison of our method with existing methods on CIFAR10-DVS and DVS128 Gesture.

| Method | CIFAR10-DVS | | DVS128 Gesture | |
|---|---|---|---|---|
| | Time step | Acc | Time step | Acc |
| LIAF-Net Wu et al. [2021] | 10 | 70.4 | 60 | 97.6 |
| TA-SNN Yao et al. [2021] | 10 | 72.0 | 60 | 98.6 |
| tdBN Zheng et al. [2021] | 10 | 67.8 | 40 | 96.9 |
| PLIF Fang et al. [2021b] | 10 | 74.8 | 20 | 97.6 |
| Spikformer Zhou et al. [2023b] | 10 | 78.6 | 10 | 95.8 |
| Spikingformer Zhou et al. [2023a] | 10 | 79.9 | 10 | 96.2 |
| **ours** | 10 | **82.67** | 10 | **97.19** |

**CIFAR10-DVS** is a neuromorphic dataset that is obtained from the CIFAR-10 dataset through a DVS camera. There are 10k images in CIFAR10-DVS, and we split them into 9k training images and 1k test images. We downsample the image resolution from $128 \times 128$ to $48 \times 48$ and input them to STMixer-2-512-8 structure SNN. We don't use the IPSPS module here because we find it does not enhance the performance on the neuromorphic dataset too much. For the training setting, we employ the AdamW optimizer with a learning rate of 0.01 and a weight decay of 0.02. The training is conducted for 300 epochs. Meanwhile, we employ the same data augmentation methods as previous works Li et al. [2021], Deng et al. [2021].

As depicted in Table. 5, our architecture achieves the best performance on CIFAR10-DVS, demonstrating the suitability of STMixer for this neuromorphic dataset.

**DVS128 Gesture** Amir et al. [2017] is a gesture recognition dataset that consists of 11 hand gesture classes performed by 29 individuals under 3 different lighting conditions. We employ the network architecture STMixer-2-512-8 and apply the same augmentation techniques from Spikingformer except cutmix. We don't use the IPSPS module here because we find it does not enhance the performance on neuromorphic dataset too much. We train the network for 200 epochs with TET loss Deng et al. [2021] and the SML method. STMixer also achieves the best performance on this dataset.

