# OpenReview forum: "Spiking Token Mixer:  An event-driven friendly Former structure for spiking neural networks"
_NeurIPS.cc/2024/Conference — NeurIPS 2024 poster_

### Official Review · Reviewer_DoAS · 2024-07-06

**Soundness:** 3
**Presentation:** 2
**Contribution:** 2
**Rating:** 7
**Confidence:** 5

**Summary:**

To address the issue where certain operators (e.g., spiking self-attention, SSA) in existing spiking Transformers cannot be executed on asynchronous neuromorphic chips, this work designs the Spiking Token Mixer (STMixer) architecture, which consists exclusively of operations supported by asynchronous scenarios including convolutional, fully connected layers, and residual paths.

**Strengths:**

This work raises a significant issue: the current spiking self-attention operators are indeed difficult to implement on asynchronous chips. Generally, designers of SNN algorithms do not often consider their hardware execution. I find it reasonable that the authors have designed algorithms with hardware limitations in mind.

**Weaknesses:**

1. The technical contribution is limited; there is no in-depth analysis of the proposed operators, and some existing work in the field is overlooked.
2. Some key points are not clearly explained, especially the section on operators.
3. The work does not follow the latest results in the field, e.g., meta-spikeformer [1], and the performance is not SOTA.

[1] Spike-driven transformer v2: Meta spiking neural network architecture inspiring the design of next-generation neuromorphic chips. In ICLR 2024.

**Questions:**

1. Some references are incorrect and cannot be accessed. LInes 91-92, 255

2. The mathematical descriptions in Section 4.3 are not rigorous and omit some crucial information. Is the matrix X composed of continuous values or is it a spiking matrix? All the dimensional changes are ignored, which is unprofessional. You cannot expect every reader to read all your references.

3. Could the authors explain the proposed STM from the perspective of linear attention? Since the LIF following Q and K functions as a kernel, why can the spiking neuron layer be directly removed in this context?

4. SML training seems crucial, but other baseline works do not appear to use this method. Therefore, is the comparison in Table 1 unfair? Why is there no comparison with Spike-driven Transformer v1/v2 in Table 1?

5. What is SDT in Table 2? It is mentioned only once in the entire text and is not explained.

6. I suggest the authors carefully check the references. If the paper has been accepted, please cite the official version instead of the arXiv version, as this is an academic convention.

---

> ### Author Rebuttal · Authors · 2024-08-06
>
> Comment 1:
>
> The technical contribution is limited...
>
> Response to comment 1:
>
> I appreciate the reviewer's feedback but respectfully disagree that the technical contribution of this paper is limited. This paper does not merely propose a structure (implant ANN former structure into SNN) to achieve the SNN SOTA performance on GPU. Instead, it designs a model suitable for asynchronous scenarios, inspiring future research to refocus on the running environments of SNNs. As mentioned in the General Response, constructing a high-performance network using operators supported by current asynchronous hardware is significantly more challenging than using those supported by synchronous hardware. Currently, advanced SNN models use multibranch spike matrix multiplication, Hadamard product operations, and others, which allow for straightforward access to global information  but difficult to execute in event-driven asynchronous settings. The STM module proposed in this article employs simple weight matrices to mix global information across token dimensions, this technique has already proven effective in works such as ANN's MLP-Mixer[1] and External Attention[2]. Additionally, STM adjusts the number of parameters and performance by employing multiple heads (i.e., multiple mixing weight matrices).
>
> Comment 2:
>
> Some key points are not clearly explained, especially the section on operators.
>
> Response to comment 2:
>
> We appreciate your feedback and have revised Section 4.3 to provide a more comprehensive explanation of the STM operator.
> Please refer to "Response to comment 5" for reviewer ESkb.
>
> Comment 3:
>
> The work does not follow the latest results in the field, e.g., meta-spikeformer [1]...
>
> Response to comment 3:
>
> The architecture of STMixer is not at odds with the meta-transformer structure; rather, it is composed of a token mixer module and a channel mixer part. We have incorporated the spike-driven transformer into Table 1 for a comparative analysis.  Although STMixer's performance on ImageNet may not surpass that of the Spike-driven transformer v2, it's important to note that due to STMixer's ultra-lightweight structure, it only consumes one-third of the energy that the Spike-driven transformer v2 does under the same parameter scale (in a synchronous scenario). As an example, Meta-SpikeFormer (T=1) consumes 7.8mJ of energy with 31.3M parameters and achieves 75.4% accuracy on ImageNet. In contrast, STmixer-8-512-16 (T=1) with 30.12M parameters achieves 73.82% ImageNet accuracy but only consumes 2.2mJ of energy.
>
> Comment 4:
>
> Some references are incorrect and cannot be accessed. LInes 91-92, 255
>
> Response to comment 4:
>
> We apologize for the oversight. We have thoroughly reviewed the references and have corrected any inaccuracies.
>
> Comment 5:
>
> The mathematical descriptions in Section 4.3 are not rigorous and omit some crucial information...
>
> Response to comment 5:
>
> Thank you for your comment. We apologize for any confusion caused by our initial presentation. To clarify, the matrix X indeed represents a spike matrix, serving as the input for Cell 1 as shown in Fig. 1A. We acknowledge the importance of providing comprehensive dimensional information and regret the omission in our previous version. To address your concerns, we have revised Section 4.3 in the manuscript to include detailed dimensional changes and additional necessary information for each variable.
>
> Comment 6:
>
> Could the authors explain the proposed STM from the perspective..
>
> Response to comment 6:
>
> Our proposed STM is analogous to the workings of the MLP-Mixer, which uses a weight matrix to mix token dimension information. Its mixing weight matrix $W_\text{STM}^h\in\mathbb{R}^{N\times N}$ can be viewed as an approximation of the attention matrix to a certain extent.  To better illustrate this point, we remove the LIF layers from Q and K. We have found that this operation does not compromise the performance of SNN (79.81% to 80%).
>
> Comment 7:
>
> SML training seems crucial, but other baseline works do not appear to use this method. Therefore, is the comparison in Table 1 unfair? Why is there no comparison with Spike-driven Transformer v1/v2 in Table 1?
>
> Response to comment 7:
>
> Thank you for your insightful comment. Our primary intention is to illustrate the potential of the SML algorithm when optimized by the STM module, which results in a significant performance boost for STMixer. We believe this indicates the potential for SNNs to achieve outstanding performance. To address your concern about fairness in comparison, we have now included the results of STMixer without SML training in Table 1. Even without SML training, STMixer still outperforms the comparison work with an accuracy $79.79\% \pm 0.19$ accuracy on CIFAR-100 and $95.96% \pm 0.04$ on CIFAR-10 for STMixer-4-384-32 (T=4). In response to your second query, we have now also included the results of Spike-driven Transformer v1/v2 in Table 1 for a more comprehensive comparison.
>
> Comment 8:
>
> What is SDT in Table 2..
>
> Response to comment 8:
>
> We apologize for the oversight. SDT in Table 2 stands for "Standard Direct Training". This refers to our baseline training pipeline, which is conducted without the use of the SMLmethod.
>
> Comment 9:
>
> I suggest the authors carefully check the references..
>
> Response to comment 9:
>
> Thank you for your suggestion. We have carefully reviewed all the references and replaced the citations of arXiv versions with the official versions of the papers where they have been officially published. We appreciate your attention to detail and adherence to academic convention.
>
>
> [1] Tolstikhin I O, Houlsby N, Kolesnikov A, et al. Mlp-mixer: An all-mlp architecture for vision[J]. Advances in neural information processing systems, 2021, 34: 24261-24272.
>
> [2] Guo M H, Liu Z N, Mu T J, et al. Beyond self-attention: External attention using two linear layers for visual tasks[J]. IEEE Transactions on Pattern Analysis and Machine Intelligence, 2022, 45(5): 5436-5447.

---

> > ### Comment · Reviewer_DoAS · 2024-08-10
> >
> > Thank you for your response. The rebuttal addresses my concerns. "What kind of SNN can be supported by neuromorphic chips" is actually always ignored by people in the field. The authors' questions and solutions are insightful. I will raise my score.

---

> > > ### Author Response · Authors · 2024-08-13
> > >
> > > Thank you for your insightful comments and your appreciation of our work.

---

### Official Review · Reviewer_ESkb · 2024-07-11

**Soundness:** 3
**Presentation:** 4
**Contribution:** 3
**Rating:** 7
**Confidence:** 5

**Summary:**

Most of the spiking neural network architectures can not truely show the superiority on the neuromporphic hardware, since in event-driven scenarios, the spike arrival times are not  precise and could result in significant differences in the output, like there is a max pooling layer. This paper propose the Spiking Token Mixer, which can well handle the problem. The authors also have validated that STMixer could achieve performance on par with or even surpasses existing Spikformer-like works in synchronous scenarios.

**Strengths:**

1. The paper is well-organized and written in a clear manner, making it accessible to readers interested in the topic.

2. The paper solves an important problem in the SNN filed and the proposed method is interesting and new.

3. The paper povide various experiments and ablations to  show the effectiveness of the proposed method.

**Weaknesses:**

1. Does the proposed method increase the energy consumption? The authors could provided detailed explanations.

2. The ablation experiments could include different types of datasets and models.

3. The comparison methods in the experiments contrasting with other approaches on CIFAR-10 may not be the most up-to-date.

**Questions:**

1.  Can you further  explain the meaning of the sentence in first paragraph of Introduction that "due to the accumulation of membrane potential in the IF neuron, spike arrival timing errors do not significantly impact subsequent layers.".
2. Could you provided a detailed explanation for the method?

---

> ### Author Rebuttal · Authors · 2024-08-06
>
> Comment 1:
>
> Does the proposed method increase the energy consumption? The authors could provided detailed explanations.
>
> Response to comment 1:
>
> TheSML method does not increase the energy consumption during the inference stage. After the training phase end, the SML method eliminates the added blocks, ensuring that the energy consumption of SNN inference remains unaffected. As demonstrated in Table 1, the STMixer does not introduce a significant additional energy cost compared to the Spikingformer. For instance, the energy consumption of STMixer-8-768-16 at T=1 is 4.45mJ, whereas the Spikingformer-8-768 at T=4 consumes 13.68mJ. Considering that energy consumption is proportional to the time step T, we can infer that an STMixer, with the same scale and time step, would have energy consumption comparable to that of a Spikingformer.
>
> Comment 2:
>
> The ablation experiments could include different types of datasets and models.
>
> Response to comment 2:
>
> We have conducted an ablation study on the CIFAR-10 dataset. The experiment results validate the effectiveness of all the designed components. The results are summarized in the table below.
>  | Modification | STMixer | IPSPS -> SPS | STM -> SSA | SML(STM)->SML(Conv) | SML->SDT |
>    | ------------ | ------- | ------------ | ---------- | ------------------- | -------- |
>    | Accuracy(%)  | 95.65   | 95.49        | 94.45      | 95.40               | 95.13    |
>
> Comment 3:
>
> The comparison methods in the experiments contrasting with other approaches on CIFAR-10 may not be the most up-to-date.
>
> Response to comment 3:
>
> Thank you for your valuable feedback. In the original manuscript, we indeed primarily compared our method with classical structure works and related studies. Recognizing the importance of keeping our work current and relevant, we have now updated our comparison in the revised manuscript to include more recent approaches. Specifically, these updates have been incorporated into Table 1. We believe that these changes will provide a more comprehensive and up-to-date comparison, addressing your concern.
>
> Comment 4:
>
> Can you further explain the meaning of the sentence in first paragraph of Introduction that "due to the accumulation of membrane potential in the IF neuron, spike arrival timing errors do not significantly impact subsequent layers.".
>
> Response to comment 4:
>
> During the SNN training phase, we convert the DVS stream in a time window into an image frame. For instance, if a position (x, y) of the DVS stream has 4 spikes in the time window, the value at position (x, y) in the image frame would be 4. This value is then input into the SNN for training. Suppose an Integrate-and-Fire (IF) neuron (threshold is 1.0 and employing soft reset) only receives input from the point (x, y), and the weight is 0.3. After receiving the input, the neuron would generate a spike by the end of this frame, leaving the residual membrane potential of 0.2. In asynchronous scenarios,  spikes in the DVS data stream enter the neuron one by one, not in frame format. Each arriving spike increases the membrane potential by 0.3, ultimately generating a spike and leaving a residual membrane potential of 0.2.  If there is a certain error in the arrival time of the spikes, as long as the number of spikes is constant, the neuron will still generate a spike and leave a residual membrane potential of 0.2 after operation.  Of course, the above scenario is the simplest example. In reality, there may be errors due to uneven spike generation, which can have some effects on subsequent layers.
> Starting from this simple example, we can infer that traditional operators such as convolution and full connection can alleviate the problem of pulse arrival error to a certain extent, as long as there is an IF neuron layer subsequently.
>
> Comment 5:
>
> Could you provided a detailed explanation for the method?
>
> Response to comment 5:
>
> Our approach is based on modifications to the Spikingformer model, specifically its SPS and SSA components. The modifications were made with two objectives in mind: 1) to adapt the model for asynchronous scenarios, and 2) to enhance the model's performance. To achieve the first objective, we eliminated the max pooling layer in the SPS and performed downsampling by modifying the stride of the convolutional layer. Concurrently, we proposed the STM module to replace the SSA module. The STM module only contains fully connected components and does not introduce multi-branch spike matrix operations.  The STM divides the input $X \in \mathbb{R}^{T\times C \times N}$ after the fully connected layer into H parts, each part $X^h  \in  \mathbb{R}^{T\times C/H\times N}$ is mixed at the token dimension through a fully convolutional matrix $W^h\in \mathbb{R}^{N\times N}$. We found that under the same training script, the performance of STM is not inferior to that of SSA. To achieve the second objective, we proposed the IPSPS method to replace the SPS module. A small portion of IPSPS's output tensor comes from the input directly transformed by a convolutional layer, thereby preserving sufficient input information. We also modified the surrogate module used by the SML method, using the STM module to construct the SML block, which significantly reduces the additional training overhead brought by the SML method.

---

> > ### Comment · Reviewer_ESkb · 2024-08-09
> > **Good rebuttal**
> >
> > Thanks for your response. My concerns have been well addressed.

---

> > > ### Author Response · Authors · 2024-08-13
> > >
> > > Thank you for your constructive feedback and for the updated score.

---

### Official Review · Reviewer_2hoo · 2024-07-12

**Soundness:** 2
**Presentation:** 3
**Contribution:** 2
**Rating:** 5
**Confidence:** 4

**Summary:**

This article examines problems with the SSA module in Spikformer in asynchronous scenarios and suggest a new module, the Spiking Token Mixing (STM) module, which consists solely of network components that cater to asynchronous environments. Besides, This article proposed the information protection spiking patch splitting (IPSPS) module to reduce information loss.

**Strengths:**

This article examines problems with the SSA module in Spikformer in asynchronous scenarios, which is a good observation. And authors intend to solve it by proposing STM and IPSPS.

**Weaknesses:**

1.From your code and Table 2 (SML->SDT). The performance improvement mainly comes from several SML blocks, which consist of several ANN layers. When considering adding residual connection and several SML blocks, the STMixer can even be seen as an ANN model (floats input is connected to the last layer through intermediate hidden ANN ). In my opinion, this is a very fatal question.
2. From the perspective of solving the asynchronous scenarios problem, the residual connection is also a problem, residual connections in the pre-activation shortcut are floats point element-wise addition, which also suffers the same problem in maxpooling and spiking matrix multiplication.

**Questions:**

1. The training cost (memory consumption and training time) of STM, SML, and STMixer should be reported compared with previous work.
2. The energy consumption of the SML block needs discussion.
3. A complete comparison of STM and SSA need to be carried out (report the results of replacing SSA in Spikingformer or Spikformer by STM block.).

**Limitations:**

See weaknesses and questions.

---

> ### Author Rebuttal · Authors · 2024-08-06
>
> Comment 1:
>
> From your code and Table 2 (SML->SDT). The performance improvement mainly comes from several SML blocks, which consist of several ANN layers. When considering adding residual connection and several SML blocks, the STMixer can even be seen as an ANN model (floats input is connected to the last layer through intermediate hidden ANN ). In my opinion, this is a very fatal question.
>
> Response to comment 1:
>
> We apologize for the confusion. The STMixer does not involve the ANN module during the inference stage. The primary function of the SML block [1] is to transmit effective gradients to the SNN's intermediate layers during back-propagation, and it does not contribute additional information to the backbone SNN during forward propagation. Consequently, the SML block does not impact the forward propagation of the backbone SNN during the inference phase, as a result, SML blocks can be removed after the completion of training. The reported accuracy in all experiments pertains to the backbone SNN, not the SML block. We have added relevant instructions to the updated manuscript.
>
>
> Comment 2:
>
> From the perspective of solving the asynchronous scenarios problem, the residual connection is also a problem, residual connections in the pre-activation shortcut are floats point element-wise addition, which also suffers the same problem in maxpooling and spiking matrix multiplication.
>
> Response to comment 2:
>
> In asynchronous scenarios, a pre-activation residual connection is not impossible to realize. Despite being termed as 'membrane shortcut' [2], the variable employed for the shortcut is the increment of the membrane potential, not its cumulative value (residual membrane potential). The hardware needs a circuit that supports floating-points values transmit to realize the membrane shortcut. Although spike arrival delay may slightly alter the output of the convolutional or fully connected layer, the output cumulative value remains constant. As a result, the sum of the increments in the shortcut from the current layer to the subsequent neuronal membrane potential will not change significantly. In other words, the spike arrival delay can affect the sequence of spike arrivals, but the sequence does not affect the summation, which has little impact on the shortcut.
>
> Comment 3:
>
> The training cost (memory consumption and training time) of STM, SML, and STMixer should be reported compared with previous work.
>
> Response to comment 3:
>
> We compared the training time and memory consumption of STMixer (T = 4) on CIFAR-100 when using either STM or SSA as the token mixer. And we also provide the training consumption of STMixer with SML. The table below summarizes the results. The training overhead for the three situations does not differ significantly, with the SSA case having the greatest memory consumption and the STM+SML case taking the longest training time. We hope this information is helpful. Please let us know if there are any other aspects you would like us to address.
>    | Case    | training time per epoch | memory consumption |
>    | ------- | ----------------------- | ------------------ |
>    | STM     | 33.27 s                  | 11,774 MB            |
>    | STM+SML | 40.04 s                  | 12,514 MB             |
>    | SSA     | 34.37 s                  | 13,994 MB            |
>
> Comment 4:
>
> The energy consumption of the SML block needs discussion.
>
> Response to comment 4:
>
> We apologize if there was any confusion caused previously. The SML block is primarily involved in the training phase of the Spiking Neural Network (SNN), but it does not participate in the inference phase. As such, while it does have an impact on the training energy consumption of the system during the training phase, it does not contribute to the energy consumption during the inference phase of the SNN. We hope this clarifies the role of the SML block in the energy consumption.
>
> Comment 5:
>
> A complete comparison of STM and SSA need to be carried out (report the results of replacing SSA in Spikingformer or Spikformer by STM block.).
>
> Response to comment 5:
>
> We use the CIFAR-100 training script provided by the Spikingformer study and merely replaced its SSA module with STM. Consequently, the accuracy of the SNN dropped from 76.25% to 75.86%. This outcome aligns with the network search results displayed in Figure 4, suggesting that while the SSA module entails higher computational load, its performance is slightly superior to that of STM. Although STM's performance is inferior to SSA's, it possesses two unique advantages—lower computational energy on synchronous hardware and suitability for asynchronous hardware. Moreover, as shown in Table 2, STM benefits more from the SML training algorithm, which enables STM to achieve a performance of 79.55%, significantly higher than SSA's 78.33%. This indicates that STM holds the potential to rival the performance of the SSA module.
>
> [1] Deng S, Lin H, Li Y, et al. Surrogate module learning: Reduce the gradient error accumulation in training spiking neural networks[C]//International Conference on Machine Learning. PMLR, 2023: 7645-7657.
> [2] Hu Y, Deng L, Wu Y, et al. Advancing spiking neural networks toward deep residual learning[J]. IEEE Transactions on Neural Networks and Learning Systems, 2024.

---

> > ### Author Response · Authors · 2024-08-13
> >
> > Dear Reviewer,
> >
> > I hope this message finds you well. We greatly appreciate the time and effort you’ve already invested in reviewing our work, and we truly value your comments. We are writing to kindly follow up on the rebuttal we submitted and sincerely hope you could review it and update the scores if your concerns have been resolved. If there are any additional points you would like us to address, we would be more than happy to provide further clarification.
> >
> > Thank you once again for your time and consideration.
> >
> > Best regards,
> >
> > Authors

---

> > > ### Comment · Reviewer_2hoo · 2024-08-13
> > >
> > > Thank you for addressing my concerns. I have updated the score.

---

> ### Author Response · Authors · 2024-08-13
>
> I would like to express our sincere gratitude for your prompt response, for taking the time to review our rebuttal, and for increasing the score.

---

### Author Rebuttal · Authors · 2024-08-06

**General Response**

We appreciate all of the reviewers' comments and reviews. Here, we would like to provide a general response to reemphasize the motivation of this paper and its contribution to the SNN field.

The main goal of this work is to design a well-performing SNN model that is friendly to asynchronous environments. Currently, SNN hardware bifurcates into synchronous and asynchronous types. Asynchronous hardware, devoid of a hardware clock and entirely event-driven, emerges as the ideal choice for SNN due to its substantially lower energy consumption. Nevertheless, it only supports very few network operations compared to synchronous hardware.

On the other hand, current advanced SNN models have adopted the design philosophy of Transformer, enhancing their performance significantly beyond previous spiking CNN models on GPU simulation. However, except for matrix addition, many of these SOTA SNN models necessitate other operations (e.g., matrix multiplication) when merging multi-branch spike matrices, which require precise timings for spike arrivals. In asynchronous environments, it is very challenging to ensure the simultaneous arrival of two spikes. Related works often overlook discussions about the SNN running environment. Our study aims to provoke researchers to rethink the adaptability of SNN models to different running environments.

Given the limited network operations supported by asynchronous hardware, the design of asynchronous friendly SNN is constrained, making the task of achieving superior performance with new models far more challenging than with existing advanced models. Consequently, the goal of this research is not to outperform the current SOTA SNN models, but to find an SNN model that is both compatible with asynchronous environments and capable of excellent performance in synchronous settings. With these considerations in mind, we propose the STMixer, which is composed of convolution, fully connected, and residual structures. It does not involve operations other than addition when merging multiple branches; thus, theoretically, STMixer supports asynchronous environments. Concurrently, our experiments demonstrate that STMixer can achieve outstanding performance even at extremely low time steps, suggesting its potential as an alternative SNN model for synchronous hardware as well.

STMixer can provide guidance for the design of asynchronous-friendly SNN models. And we hope that  this paper will guide future SNN structural design efforts to pay more attention to implanting environment issues, further yielding a greater number of excellent asynchronous-friendly SNN models and training algorithms.

---

### Decision · Program_Chairs · 2024-09-25

**Decision:**

Accept (poster)

**Comment:**

The authors argue that certain operators used in Transformer implementations are not very suitable for asynchronous spiking based neuromorphic chips. The authors make an attempt to provide ways so that the self-attention module can be implemented in a way suitable for asynchronous chips. The reviewers agree that that the authors are focusing on a real problem that limits the applicability of transformers on asynchronous chips.

Among the weaknesses listed are presentation issues (missing or incorrect references and notational issues making some parts of the paper feel a bit unprofessional). Another weakness is that the authors seem to make claims on the suitability of the algorithm for implementation on neuromorphic hardware, yet they do not seem to take into account issues that may hinder the adoption of such an algorithm on actual hardware. These can include issues that are specific to each hardware implementation, such as the fan-out supported by the hardware's neurons or the spike rate supported.

Nevertheless there is consensus among the reviewers that the merits of the paper outweigh the drawbacks, and the authors seem to have answered the reviewer questions to their satisfaction. The authors should fix all the presentation issues and the references issues mentioned by the reviewers before publication.